# Application of Convolutional Neural Networks for Diagnosis of Eosinophilic Esophagitis Based on Endoscopic Imaging

**DOI:** 10.3390/jcm11092529

**Published:** 2022-04-30

**Authors:** Eiko Okimoto, Norihisa Ishimura, Kyoichi Adachi, Yoshikazu Kinoshita, Shunji Ishihara, Tomohiro Tada

**Affiliations:** 1Department of Internal Medicine II, Shimane University Faculty of Medicine, Izumo 693-8501, Japan; okimotoeiko@outlook.com (E.O.); si360405@med.shimane-u.ac.jp (S.I.); 2Health Center, Shimane Environment and Health Public Corporation, Matsue 690-0012, Japan; adachi@kanhokou.or.jp; 3Department of Medicine, Hyogo Prefectural Harima-Himeji General Medical Center, Himeji 670-8560, Japan; kinositamove2@yahoo.co.jp; 4AI Medical Service Inc., Toshima, Tokyo 170-0013, Japan; tadatomo@ai-ms.com

**Keywords:** artificial intelligence, convolutional neural network, eosinophilic esophagitis, endoscopy

## Abstract

Subjective symptoms associated with eosinophilic esophagitis (EoE), such as dysphagia, are not specific, thus the endoscopic identification of suggestive EoE findings is quite important for facilitating endoscopic biopsy sampling. However, poor inter-observer agreement among endoscopists regarding diagnosis has become a complicated issue, especially with inexperienced practitioners. Therefore, we constructed a computer-assisted diagnosis (CAD) system using a convolutional neural network (CNN) and evaluated its performance as a diagnostic utility. A CNN-based CAD system was developed based on ResNet50 architecture. The CNN was trained using a total of 1192 characteristic endoscopic images of 108 patients histologically proven to be in an active phase of EoE (≥15 eosinophils per high power field) as well as 1192 normal esophagus images. To evaluate diagnostic accuracy, an independent test set of 756 endoscopic images from 35 patients with EoE and 96 subjects with a normal esophagus was examined with the constructed CNN. The CNN correctly diagnosed EoE in 94.7% using a diagnosis per image analysis, with an overall sensitivity of 90.8% and specificity of 96.6%. For each case, the CNN correctly diagnosed 37 of 39 EoE cases with overall sensitivity and specificity of 94.9% and 99.0%, respectively. These findings indicate the usefulness of CNN for diagnosing EoE, especially for aiding inexperienced endoscopists during medical check-up screening.

## 1. Introduction

Over the past two decades, eosinophilic esophagitis (EoE) has been increasingly recognized worldwide, especially in Western countries [1,2]. In Asia, including Japan, while it has been noted as a rare condition, prevalence has rapidly risen in recent years [3,4,5,6]. EoE is a clinicopathological condition characterized by symptoms, typical endoscopic findings, and dense esophageal eosinophilia [7,8]. Common symptoms include dysphagia, heartburn, and chest pain, which are non-specific and can overlap those known to be associated with other esophageal diseases, such as gastroesophageal reflux disease, achalasia, and esophageal carcinoma. Therefore, distinguishing between EoE and other such diseases based on symptoms is difficult and recognition of specific endoscopic findings is a key to precise diagnosis [9]. We previously reported that linear furrows can be found in approximately 90% of EoE patients and may be an important clue for accurate diagnosis [10]. However, recognition of abnormalities requires training and experience. In our previous study, we examined the overall inter-observer agreement for a diagnosis of EoE based on characteristic endoscopic findings [11]. The kappa coefficient of reliability for a cohort of 40 endoscopists for diagnosis of EoE was 0.34 [95% confidence interval (CI): 0.33–0.35], indicating a low level of inter-observer agreement. In addition, the kappa value was significantly lower for endoscopists without board certification from the Japanese Gastroenterological Endoscopy Society as compared with those who were board-certified, suggesting that endoscopic diagnosis of EoE is sometimes difficult, especially for inexperienced endoscopists, and indicates a need for a computer-assisted diagnosis (CAD) system.

Recent emerging evidence has highlighted application of artificial intelligence (AI) as part of a CAD system for gastrointestinal diseases [12]. We have developed a convolutional neural network (CNN) system with an architecture for deep learning in the field of medical image analysis, and demonstrated that it could be used as a CAD system with high accuracy for diagnosis of esophageal carcinoma [13,14,15,16]. Here, we sought to establish such an AI-based diagnostic system for EoE using endoscopic images and evaluate its ability as a diagnostic utility.

## 2. Materials and Methods

### 2.1. Preparation of Training and Validation Imaging Sets

We retrospectively identified 143 patients with EoE and 331 subjects without any esophageal lesions by reviewing clinical data from individuals who had undergone esophagogastroduodenoscopy (EGD) examinations from 2013 to 2018 at Shimane University Hospital and Health Center, Shimane Environment and Health Public Corporation. Most of the subjects with normal esophageal images by EGD visited our institutes for an annual medical check-up examination. Subjects with esophageal symptoms, treated with a proton pump inhibitor (PPI) and/or steroid, and those with current or past history of esophageal diseases, such as reflux esophagitis and esophageal cancer, were excluded for preparing normal esophageal images. For those EGD examinations, white light imaging (WLI) was captured using a standard endoscope, including EVIS GIF-H290Z, GIF-H260, GIF-H260Z, and GIF-XP260N (Olympus Medical System, Co., Ltd., Tokyo, Japan), or an EG-L580NW endoscope (LASEREO endoscopic system; Fujifilm, Tokyo, Japan). EoE was diagnosed according to current clinical guidelines [7,8], which include symptoms of esophageal dysfunction and esophageal biopsy findings [(≥15 eosinophils/high power field (HPF)]. All obtained biopsy specimens were examined by certified pathologists at Shimane University Hospital. Patients in an inactive phase of EoE after treatment as well as those with a concomitant esophageal disease, such as reflux esophagitis, esophageal varices, or esophageal cancer, were excluded from analysis.

For the training image dataset used to produce the CNN algorithm, 1192 images from 165 endoscopic sessions in 108 patients with EoE and 1192 images from 235 subjects who did not show any evidence of an abnormal lesion in the esophagus were used. For the CNN algorithm validation image dataset, an additional 249 images from 39 endoscopic sessions in 35 patients with EoE, and 507 images from 96 subjects who did not have either esophageal symptoms or abnormal endoscopic findings in the esophagus were utilized (Figure 1). Two expert endoscopists (E.O., and N.I.) selected all images of patients with EoE that showed at least one of the following endoscopic findings; linear furrows, whitish exudates, rings, edema, or stricture, which are included in the endoscopic reference score (EREFS), and the total EREFS was calculated [17]. Images with low quality resulting from halation, motion-blurring, defocus, or excessive mucus were excluded from this study. Representative endoscopic images showing EoE and normal esophagus findings are presented in Figure 2. Images obtained by image enhanced endoscopy (IEE), such as narrow band imaging (NBI), blue laser imaging (BLI), or magnification endoscopy, were not included.

The study protocol was evaluated and approved by the Ethical Committee of the Shimane University Faculty of Medicine (No. 3426) and the Health Center, Shimane Environment and Health Public Corporation.

### 2.2. Convolutional Neural Network Algorithm

To construct an AI-based diagnostic system, we used the ResNet50 deep neural network architecture (https://arxiv.org/abs/1512.03385 accessed on 24 February 2022), which is comprised of 50 layers for image classification. The CAD system was trained for 100 epochs with a learning rate of 0.0001 and a stochastic gradient descent optimization function. Selected images were randomly rotated to orientations of 0, 90, 180, and 270 degrees for data augmentation to improve the accuracy of the model trained by CNN.

### 2.3. Outcome Measures and Statistics

To set an optimal cut-off value for the probability score to detect EoE, a receiver operating characteristic (ROC) curve was drawn, and sensitivity and specificity were evaluated using Youden’s index (sensitivity + specificity − 1). Youden’s index is one of the standard methods for determining the most ideal cutoff value using sensitivity and specificity. When the probability score was higher than the cut-off value, the image was regarded as positive for EoE findings.

To predict whether the case could be diagnosed as EoE, two criteria were used. Criterion A indicated such a diagnosis if at least one image was shown positive for EoE by AI and criterion B indicated that when more than half of the images were positive for EoE by AI. A sensitivity, specificity, accuracy, positive predictive value (PPV), and negative predictive value (NPV) of AI to diagnose EoE were calculated.

## 3. Results

### 3.1. Clinical Characteristics of Patients in Training and Validation Image Dataset

Training image dataset were obtained from 165 endoscopic sessions of 108 EoE patients in an active phase that showed eosinophilic infiltration of more than 15 per HPF. Validation image datasets were obtained from an additional 39 endoscopic sessions of 35 EoE patients in an active phase. Demographic and clinical characteristics of those patients are shown in Table 1. The patients in validation dataset consisted of 30 males and 5 females, with a mean (± SD) age of 46.9 ± 10.0 years (range 36–84 years). Twenty-five (71.4%) had a concurrent allergic disease, with allergic rhinitis the most frequent. As for endoscopic findings, edema, linear furrows, rings, and whitish exudates were frequently observed, while edema was found in all cases. No stricture was found in any of enrolled patients. The total score of EREFS per EGD (median) was 3 (range, 1–6). These characteristic endoscopic findings were consistent with those noted in previous studies conducted in Japan [9,10].

### 3.2. AI-Based Diagnosis of Each Image

Based on ROC curve analysis, 0.995 was set as the optimal cut-off value for probability score (Figure 3). The accuracy, sensitivity, specificity, PPV, and NPV of the AI-based diagnosis for EoE were 94.7%, 90.8%, 96.6%, 93.0%, and 95.5%, respectively (Table 2). Images were obtained using transoral (*n* = 574) or transnasal (*n* = 182) endoscopy, and accuracy was similar between those procedures (93.7% [95% CI 91.4–95.6] and 97.8% [95% CI 94.5–99.4], respectively). The trained CNN evaluated the 756 images in the validation set at a speed of 21 s.

### 3.3. AI-Based Diagnosis of Each Case

The median number of images per patients was six (range 2–22). Using criterion A, i.e., EoE determined when at least one image was positive for EoE findings, accuracy, sensitivity, specificity, PPV, and NPV were 88.1%, 97.4%, 84.4%, 71.7%, and 98.8%, respectively. When criterion B was used, i.e., EoE determined when more than half of the images were positive for EoE findings, accuracy, sensitivity, specificity, PPV, and NPV were 97.8%, 94.9%, 99.0%, 97.4%, and 97.9%, respectively. Thus, these results suggested that EoE was accurately diagnosed by AI using criterion B (Table 3). Indeed, with criterion B, 37 of the 39 of the EoE cases in the validation set were correctly diagnosed as EoE.

### 3.4. Causes for False Positives and False Negatives

Potential factors that can cause false positive or negative results in AI-based diagnosis are summarized in order of frequency in Table 4. Of 507 normal esophageal images, 17 were misdiagnosed as EoE. More than half of the false-positive results were considered to be caused by misdiagnosis of normal structures, such as esophageal vertical folds or transient concentric rings, as EoE (Figure 4A,B). Of 249 images of EoE cases, 23 were misdiagnosed as normal, with the most frequent (n = 13, 56.5%) cause of false-negative results considered to be a minor endoscopic finding termed the Ankylosaurus back sign (ABS) with edema [18,19]. One of the false-negative cases showed ABS with edema without other major endoscopic features (Figure 4C). In addition, the present AI-based system was not able to diagnose mild or obscure findings in 10 images, even when a major feature of EoE such as linear furrows was present (Figure 4D).

## 4. Discussion

Due to the absence of reliable EoE diagnostic biomarkers [20,21], endoscopy with biopsy is the current gold standard modality for detecting and diagnosing esophageal eosinophilia, which is an essential finding for determining EoE. Moreover, because the distribution of esophageal eosinophilia is considered to be patchy and uneven [22,23], current clinical guidelines recommend that at least six biopsy samples should be obtained from different locations, with the focus on areas with endoscopic mucosal abnormalities, since diagnostic sensitivity increases with the number of samples and is maximized after obtaining at least six [7,8]. On the other hand, repeated biopsy procedures are costly and time-consuming, and severe adverse effects such as bleeding may occur, especially in patients undergoing anti-coagulation therapy. Characteristic endoscopic features for EoE include linear furrows, rings, whitish exudates, edema, and stricture, with at least one of these abnormalities reported to be detected in over 90% of examined EoE patients [24]. In addition, eosinophilic peak counts were significantly higher in areas of the esophagus with characteristic endoscopic features, such as linear furrows and whitish exudate, as compared to normal-appearing areas [10,25]. However, these features are not specific for EoE, and inter-observer agreement among Japanese endoscopists regarding endoscopic diagnosis of EoE has been reported to be low [11]. Therefore, it can be challenging to diagnose EoE, because the endoscopic features are subtle and easily missed, especially by an inexperienced endoscopist. In addition, diagnostic delay may lead to esophageal stenosis and impair patient quality of life [26].

To overcome interobserver variability and learning curve issues, AI using CAD based on deep learning has recently been introduced to assist endoscopists with the detection and diagnosis of upper gastrointestinal diseases, including not only neoplastic but also benign inflammatory diseases, such as *Helicobacter pylori*-associated atrophic gastritis [27]. Recently, Guimaraes et al. reported a novel AI system to distinguish EoE from esophageal candidiasis and normal esophagus [28]. In that study, a small number of patients with EoE (*n* = 7) was used for the test dataset, but the CNN-based algorithm for the global diagnosis demonstrated significantly higher accuracy (91.5%), sensitivity (87.1%), and specificity (93.6%), as compared with those by endoscopists. We sought to establish a CNN algorithm for diagnosis of EoE in Japanese patients based on the analysis of endoscopic images by both transoral and transnasal endoscopy, since endoscopic characteristics of EoE have been reported to be different from those in Western countries [4,18]. Consistently, the present results indicated that the system can diagnose EoE with high accuracy in a short period of time.

WLI was evaluated in this study, whereas IEE, such as NBI, BLI, and magnification endoscopy, was not. Peery et al. reported fair to good interobserver agreement regarding endoscopic findings of rings and furrows, but poor agreement regarding plaques, and noted that agreement did not improve with the addition of NBI to WLI. In contrast, NBI with magnification endoscopy was found to be a reliable diagnostic modality for EoE [29,30], though that is not generally used for healthy subjects as part of a medical check-up. Additional investigation is needed to evaluate whether non-magnifying NBI observation is useful for detection of EoE with an AI system.

The present study included archived images obtained by transnasal endoscopy, which have less information as compared to those obtained by high definition transoral endoscopy [31]. In Japan, most cases of EoE are diagnosed based on screening transnasal endoscopy examinations performed as part of a medical check-up [25,32]. The majority of those with esophageal eosinophilia are asymptomatic or have only faint symptoms, thus the affected individual generally does not go to a hospital for a more detailed examination, while endoscopic findings between EoE and asymptomatic esophageal eosinophilia cases are not different [32,33,34]. Under the assumption that an asymptomatic subject with typical endoscopic findings and esophageal eosinophilia is in an early stage of EoE [35], endoscopy performed at a medical check-up examination may be a good opportunity for early diagnosis of EoE. Our CAD system may assist endoscopists in making the positive diagnosis of EoE.

Some of the present cases had false-negative results, more than half of which were due to minor endoscopic findings. We recently reported a novel endoscopic finding termed Ankylosaurus back sign (ABS) in patients with EoE [19]. Although it was less frequent in EoE cases than typical endoscopic features such as linear furrows, it was closely associated with response to proton pump inhibitor treatment in patients with EoE [19]. AI with deep learning developed by including minor features such as ABS, in addition to EREFS, will further improve its accuracy for diagnosis.

Among characteristic endoscopic findings, stricture was not included in this study, as that is a very rare finding in Japanese patients with EoE. A stricture is easy to recognize during an endoscopy examination. This study aimed to diagnose EoE at an early stage in individuals with mild or no symptoms who were undergoing a medical check-up so as to avoid a delay in diagnosis, which could lead to the development of an esophageal stricture. The results demonstrated that the present system can assist an endoscopist with diagnosis of EoE in the absence of severe complications.

This study has several limitations. First, still images from only EoE patients and healthy subjects were retrospectively examined, and the number was small. It will be necessary to train the system using video with other disease datasets such as esophageal carcinoma before activating it for clinical use. Second, endoscopic images with normal appearing mucosa were selected without histological confirmation with a biopsy sample. Although subjects treated with PPI and/or steroid, and those with current or past history of esophageal diseases were strictly excluded for preparing normal esophageal images, selection bias may affect the results. Furthermore, subjects with esophageal symptoms and/or any abnormal endoscopic findings in the esophagus were excluded, though up to 10% of EoE patients have been reported to show normal appearing mucosa [24,36]. Finally, for diagnosis of EoE using criterion B, the ratio of images with EoE findings among all images might have been affected by other lesions and/or conditions in the esophagus. Therefore, criteria used to determine a case as positive for EoE should be modified for use in clinical settings.

## 5. Conclusions

The present CNN system used to analyze multiple esophageal images demonstrated high sensitivity, specificity, and accuracy for detection of EoE within a very short time period. These results show the usefulness of this system for diagnosis of EoE and also indicate that it may be especially helpful for inexperienced endoscopists during medical check-up screening.

## Figures and Tables

**Figure 1 jcm-11-02529-f001:**
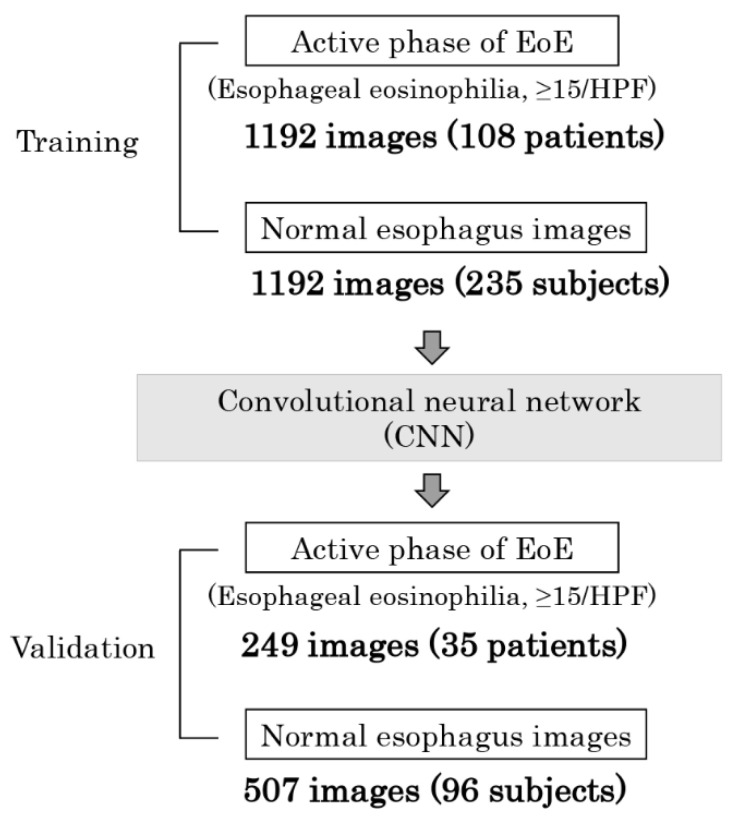
Flow diagram of CNN learning.

**Figure 2 jcm-11-02529-f002:**
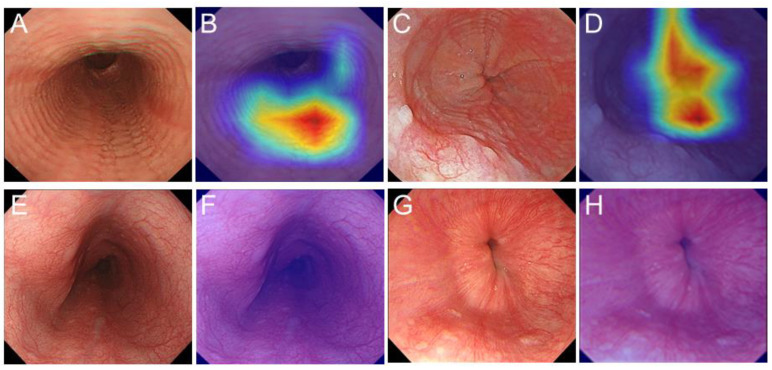
Representative cases of correct EoE detection with artificial intelligence (AI)-based diagnosis system. (**A**) Case of EoE showing edema, linear furrows, and rings in the middle esophagus. (**B)** Images overlaid with relevance heatmaps (probability score; 1). (**C**) Case of EoE showing linear furrows and whitish exudates in the lower esophagus. (**D**) Images overlaid with relevance heatmaps (probability score; 1). (**E**,**G**) Subject with normal esophagus. (**F**,**H**) Images overlaid with relevance heatmaps (probability score; <0.01).

**Figure 3 jcm-11-02529-f003:**
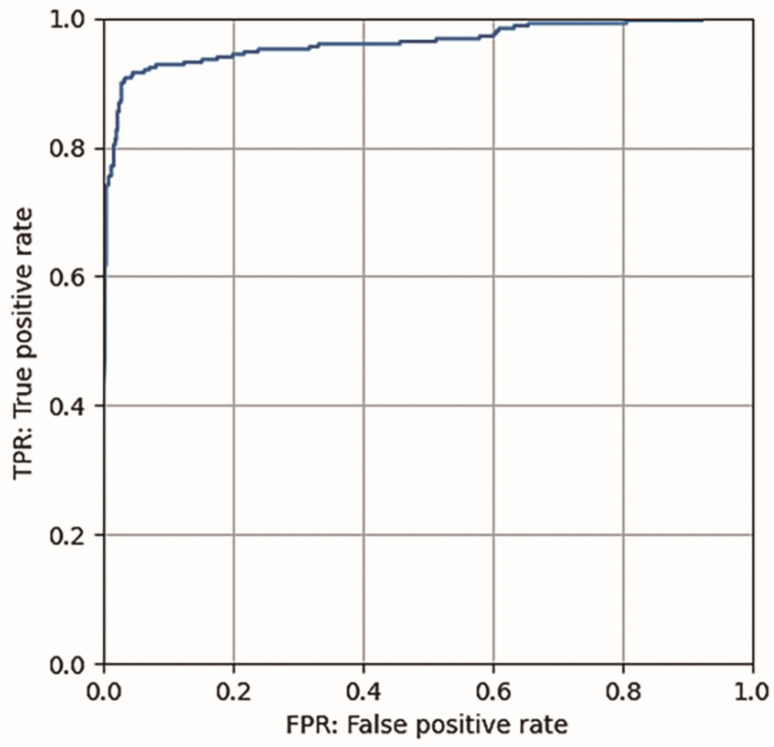
Receiver-operating characteristic curve.

**Figure 4 jcm-11-02529-f004:**
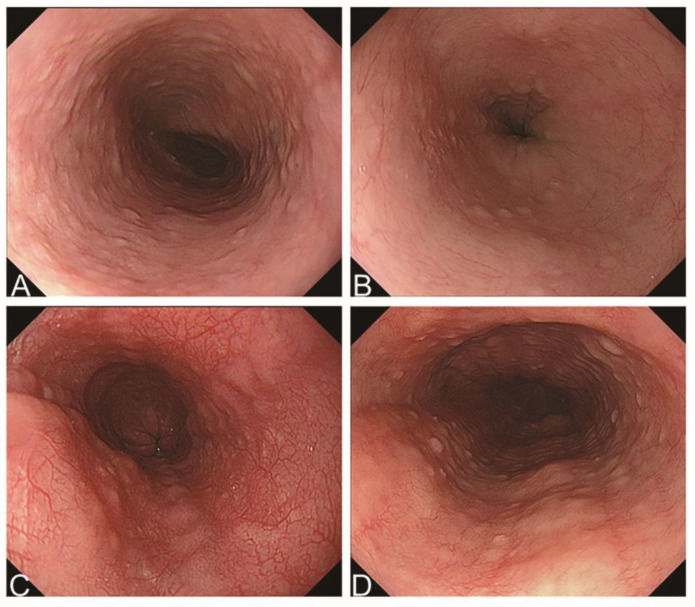
Sample false-positive and false-negative images. (**A**,**B**) Misdiagnosis of normal structures. (**C**) Ankylosaurus back sign. (**D**) Obscure findings related to linear furrows.

**Table 1 jcm-11-02529-t001:** Clinical characteristics of enrolled patients in training and validation study.

	Training Dataset	Validation Dataset
Patient characteristics	*n* =108	*n* = 35
Male, no. (%)	89 (82.4)	30 (85.7)
Age, years, mean (SD)	48.4 (11.6)	46.9 (10.0)
Concurrent allergic disease, no. (%)	80 (74.1)	25 (71.4)
Allergic rhinitis	52 (48.1)	14 (40.0)
Bronchial asthma	22 (20.4)	9 (25.7)
Atopic dermatitis	20 (18.5)	6 (17.1)
Symptom, no. (%)		
Dysphagia	62 (57.4)	20 (57.1)
Heartburn/regurgitation	42 (38.9)	16 (45.7)
Endoscopic characteristic	*n* = 165	*n* = 39
Edema, no. (%)	162 (98.2)	39 (100)
Linear furrows, no. (%)	131 (79.4)	28 (71.8)
Rings, no. (%)	96 (58.2)	27 (69.2)
Whitish exudates, no. (%)	106 (64.2)	23 (59.0)
Stricture, no. (%)	0 (0)	0 (0)
Ankylosaurus back sign, no. (%)	28 (17.0)	6 (15.4)
EREFS (total score), median (range)	3 (1–6)	3 (1–6)

EREFS, endoscopic reference score.

**Table 2 jcm-11-02529-t002:** Diagnostic accuracy for each image.

Accuracy	94.7 (92.9–96.2)
Sensitivity	90.8 (86.5–94.1)
Specificity	96.6 (94.7–98.1)
PPV	93.0 (89.0–95.9)
NPV	95.5 (93.3–97.1)

Data are presented as % (95% confidence interval). NPV, negative predictive value; PPV, positive predictive value.

**Table 3 jcm-11-02529-t003:** Diagnostic accuracy for each case.

	Criterion A	Criterion B
Accuracy	88.1 (81.4–93.1)	97.8 (93.6–99.5)
Sensitivity	97.4 86.5–99.9)	94.9 (82.7–99.4)
Specificity	84.4 (75.5–91.0)	99.0 (93.7–100.0)
PPV	71.7 (57.7–83.2)	97.4 (84.9–100.0)
NPV	98.8 (93.4–100.0)	97.9 (92.7–99.7)

Data are presented as % (95% confidence interval). NPV, negative predictive value; PPV, positive predictive value.

**Table 4 jcm-11-02529-t004:** Causes of false-positive and -negative results in CNN diagnosis.

False-Positive (*n* = 17)	No. of Images (%)
Normal structure (vertical fold/transient concentric rings/glycogenic acanthosis/EGJ)	6/3/2/1 (58.8)
Influence of light (shadow)	4 (23.5)
Whitish deposit	1 (5.9)
False-negative (*n* = 23)	No. of images (%)
Minor endoscopic finding (Ankylosaurus back sign)	13 (56.5)
Obscure lesion (linear furrows/rings/whitish exudates)	5/2/2 (39.1)
Unknown	1 (4.3)

EGJ, esophagogastric junction.

## Data Availability

The data presented in this study are available on reasonable request from the corresponding author.

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
