# Peer review of "Application of Convolutional Neural Networks for Diagnosis of Eosinophilic Esophagitis Based on Endoscopic Imaging"

_jcm, 2022, doi:10.3390/jcm11092529_

Round 1
Reviewer 1 Report
Congratulations to the authors on this exciting study. Okimoto et al. evaluated the application of neural networks for helping diagnose eosinophilic esophagitis.
Overall, the study has an appropriate design and is well-written.
I have some comments:
- The authors should clearly state that it is a case-control study. This information impacts in the quality of evidence and risk of bias of diagnostic studies.
- The correct Youden test is:
- The pretest chance needs to be more detailed. How was the control group chosen? Why were these patients submitted to endoscopy? What was the indication for the endoscopy investigation?
Reviewer 2 Report
I congratulate the authors for their esteemed efforts that resulted in a well documented, original, and convincing research.Before going further with the paper, I have some comments:
In the results section, it would be fairer to present the clinical characteristics of the training set, also. Moreover, it would be more scientifically sound to present clinical characteristics for each group (EoE vs healthy) and test whether there are significant differences between each variable/characteristic. For example, if the groups would differ regarding concurrent alergic diseases, this may introduce a possible bias (which eventually should be discussed).
I invite the authors to discuss more on how they see this CAD system used in practice, as it is not clear from the paper. The plan is to use it to make positive diagnosis or to help the clinician easily identify the best esophageal regions for biopsy sampling?
Attention to this phrase: "... upper gastrointestinal diseases, including not only neoplastic but also benign inflammatory diseases, such as Helicobacter pylori-associated atrophic gastritis and ulcerative colitis". Ulcerative colitis is not an upper gastrointestinal disease.
In this phrase the verb seems to miss: "In that study, small number of patients with EoE (n=7) was used for test dataset, but CNN-based algorithm for the global diagnosis with significantly higher accuracy (91.5%), sensitivity (87.1%), and 206 specificity (93.6%) as compared with those by endoscopists."
This phrase needs a reference or it can be removed as its relevance is not clear: "Although that was less frequent in EoE cases than typical endoscopic features such as linear furrows, it was closely associated with response to proton pump inhibitor treatment in patients with EoE"
Round 2
Reviewer 2 Report
Congratulations to the authors!
I checked the revised manuscript and all my comments have been answered accordingly. I have no further observations.
Good luck!